# The Burden of Omicron Variant in Pakistan: An Updated Review

**Sarmir Khan** [1,*], **Samra Hayat Khan** [2], **Fatima Haider** [2], **Jaweria Malik** [2], **Feroz Khan** [3], **Ikram Khan** [4], **Ihsan Ullah** [5], **Muhammad Amir Zia** [1] **and Shazia Kousar** [6,*]

1. National Institute for Genomics and Advanced Biotechnology (NIGAB), National Agricultural Research Center (NARC), Islamabad 44000, Pakistan
2. Department of Bioinformatics and Biosciences, Capital University of Science and Technology, Islamabad 44000, Pakistan
3. Department of Zoology, Wild Life and Fisheries, PirMehr Ali Shah Arid Agriculture University, Rawalpindi 46000, Pakistan
4. School of Life Science, Lanzhou University, Lanzhou 730000, China
5. Oilseeds Program, National Agricultural Research Center (NARC), Islamabad 44000, Pakistan
6. Dr. Ruth KM Pfau Civil Hospital, Msson Rd., Karachi 74200, Pakistan
* Correspondence: sarmirkhan999@gmail.com (S.K.); shaziakousar110@gmail.com (S.K.)

**Abstract:** A new COVID-19 variant, Omicron, has emerged from South Africa, indicating that the pandemic will not be over anytime soon. An unimaginable amount of economic damage has resulted from the pandemic. Omicron and its economic implications are discussed in detail in this review article. It also includes statistics on occurrence, mortality, and recuperation in Pakistan and data on the country's immunization coverage. Developing countries with unstable economies, such as Pakistan, have experienced increased economic difficulties. As a result, all developed and underdeveloped countries should strictly adhere to preventive measures and improvements in diagnostic tests and vaccine development to bring the pandemic under control.

**Keywords:** COVID-19; Omicron; pandemic; Pakistan; vaccine

## 1. Introduction

As the Omicron variant emerged, it seems as if the pandemic is not ending anytime soon. The Omicron variant has up to 32 mutations. Thus, it is highly divergent, making it more difficult to comprehend than COVID-19's parent variants [1]. Emerging from the South African region and categorized as the "Variant of Concern" [2], the World Health Organization (WHO) has identified the variation in COVID-19's genetic architecture as having such a high impact that it will greatly affect the course of the pandemic [3]. Moreover, strong evidence suggests that the rate of transmissibility for the Omicron variant is comparatively much higher than any variants that have emerged until now. Still, on the contrary, it yields to be a "milder" version, the proof of which is reduced hospitalizations [2]. However, WHO soon declared that the reduced need for hospitalizations and fewer death rates does not in any way mean that the variant is "milder", rather the real-time evolution analysis of the variant shows that it even has "the properties of immune escape", thus, proving it to be dangerous [4]. In terms of symptoms, the World Health Organization's spokesperson, Dr. Maria Van Kerkhove, reported that in the analysis of the variant, Omicron, it has been established that the variant does not bring about any new change to the "disease profile" [5]. That is to say that, to date, there have been no new symptoms, other than the previous variants' symptoms reported with the emergence of the Omicron.

On the other hand, with the latest wave carried forward by the Omicron variant, the economy has suffered more damage that has taken over the previous damage, of which had not been fully catered for. In short, the economy's recovery that was gradually

building its way has been disrupted by the Omicron variant [6]. Casselman and Ember (2022) elaborate that the advent of the Omicron variant is the cause of "more missing workers, more uncertainty, and higher inflation", thus, impacting the economy with a more violent force.

The current article investigates and focuses on all associated information of the variant from its history, along with transmissive efficiency prevention, all the way up to its treatment. Follow the article for a more elaborative overview of the Omicron Variant, its history, symptoms, prevention, impact on the economy, and treatment.

**Sars-CoV-2 Pathogenesis**

Four structural proteins comprise the SARS-CoV-2 virion: the Envelope Protein (E), the Membrane Protein (M), the Nucleocapsid Protein (N), and the Spike Protein (S). SARS-CoV-2 induces infection via four stages of development. During the early stages, angiotensin-converting enzyme 2 (ACE2) binds to the S1 motif of the host cell's S protein, resulting in the cleavage of the S protein by TMPRSS2 and activation of the S2 motif in anticipation of the stage 3 fusion. When S2 stimulation is present at stage 4, the proactive, single-stranded RNA molecule of a virus is more likely to be incorporated into the recipient's DNA. Initiation of MDA5 or RIG-I in response to DNA precursor replication may initiate a chain reaction through MAVS, leading to the generation of type I and type III interferons (IFNs). Interferons induce interferon-stimulated gene expression by activating paracrine and autocrine signaling channels through plasma membrane receptors and the JAK–STAT1/2 signaling complex (Figure 1) [7].

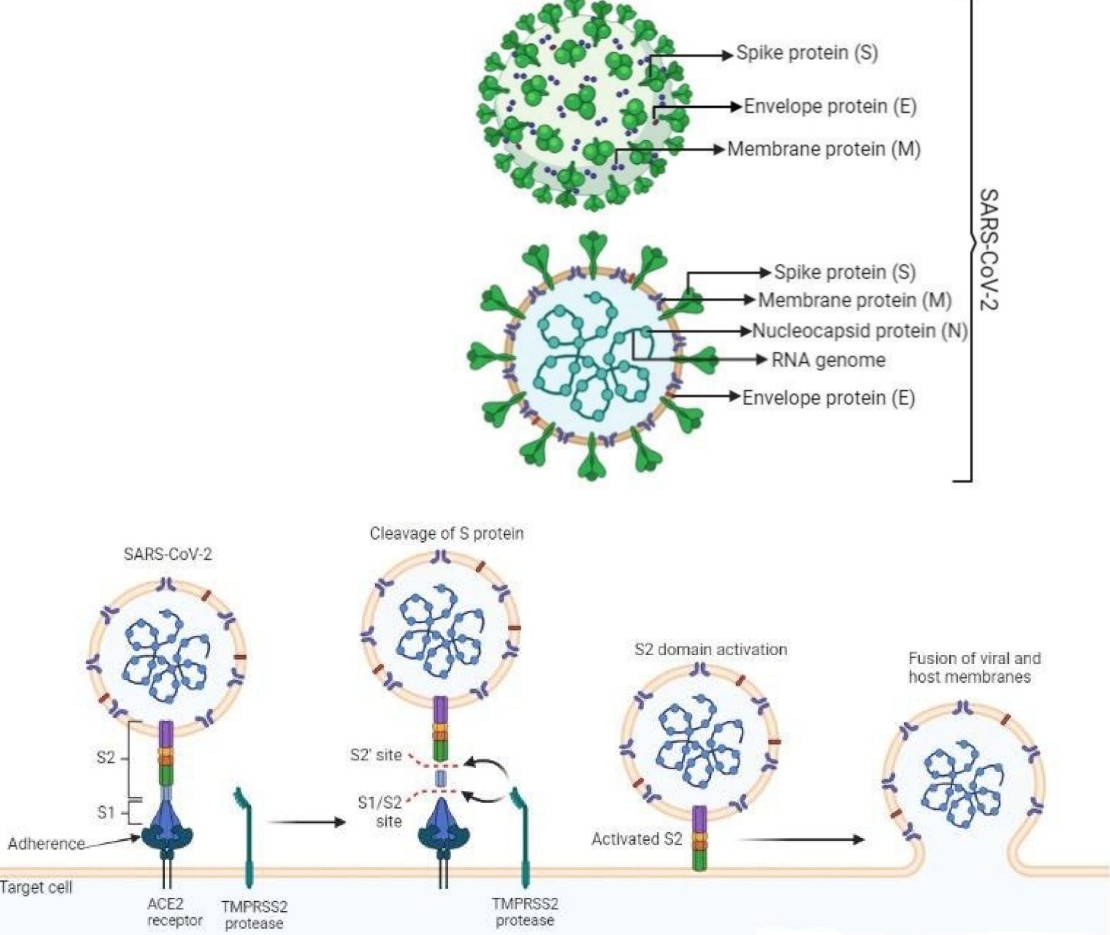

**Figure 1.** *Cont.*

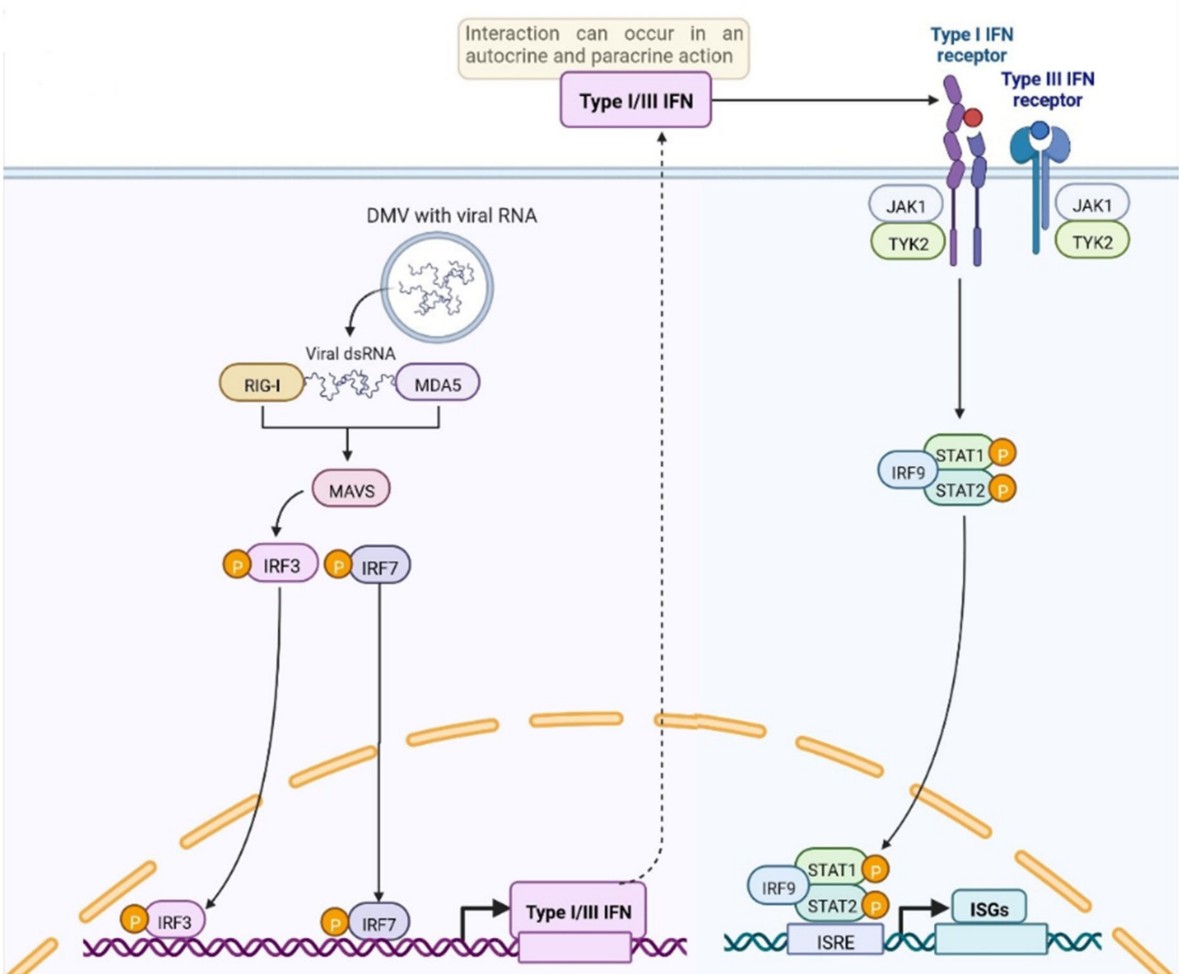

**Figure 1.** Pathogenesis of SARS-CoV-2.

### 1.1. Genomics Properties of Omicron Variant

There has been a discovery of 18,216 modifications in the genomic architecture of the Omicron version of SARS-CoV-2 with more than 97 percent of them occurring in the coding area and 558 occurring in the extragenic region. More than 97 percent of these mutations in the coding area have been discovered in the Omicron variant's genome. There are 11,995 synonymous single-nucleotide polymorphisms, 2743 synonymous single-nucleotide mutations among the 2965 indels, and no synonymous mutations in the coding region [8]. The detection of 30 mutations in the spike proteins has been done, out of which most were present in the receptor-binding region [9]. Having a high number of mutations on spike proteins makes it transmissible and allows it to partially evade even vaccines. Due to over 270 million SARS-CoV-2 infections all around the globe, there has been an evolution of the virus into over 1500 unique Pango line ages [10]. Scientists expect that vaccines will be less effective against the variant [11]. There is one insertion mutation and three additional deletions external to the spike protein. Their cords from GISAID revealed six deletions, 11 mutations, and one inclusion, with the variants in s 214 EPE existing distinctively [12]. Surprisingly, several of the mutations that aid in antibody neutralization was discovered in the previous VOC [13]. At various stages, five distinct SARS-CoV-2 types were considered VOC. South African epidemiologists covered mutational data indicating several of the worrisome alterations (N501Y, D614G, K417N, and T478K), as well as additional mutations in Omicron increasing the risk of partial resistance to the vaccines and also of reinfection. Out of the 3RBD mutations, two are shared by the Omicron and Delta variant. For the first one, there is a substitution of lysine to asparagine at 417 positions which has been related to the S protein structural changes and can also be responsible for improving immune evasion.

Meanwhile, on the 478 positions, there is a threonine to lysine substitution in the second mutation, which results in increasing the chance of steric obstruction and electrostatic potential of the residue. This permits immunological escape and also enhances the binding affinity of RDB [14].

Wuhan-Hu-1 includes 1273 amino acids, whereas the Delta and Omicron variants have 1271 and 1270 amino acids, respectively. Both contain fewer repetitions than the wild-type protein due to sequence deletion [15]. According to a study by Kandeel et al., the Omicron version of SARS-CoV-2 represents a novel monophyletic clade [16]. Wang et al., on the other hand, demonstrated that the Omicron version of SARS-CoV-2 developed from the 20B clade and split into double subclades [17]. The Lambda variant was first introduced in Peru in 2020. In 2020, it affected various countries in summer but no circulation of this variant was reported around the globe. This variation has seven alterations in the spike protein of the virus in comparison with previous strains of SARS-CoV-2. Some of these mutations could increase the transmissibility of the virus or decrease the capability of antibodies to neutralize the virus. According to Public Health England, the variant did not show any evidence of life-threatening disease or a reduction in the effectiveness of the vaccine. Similarly, mu is another variation of SARS-CoV-2 identified in 2021 in Columbia [18].

Genetic Variation Indistinct Omicron Variants

The most frequent alterations in the three first Omicron genotypes are D796Y, D614G, E484A, G142D, G339D, H655Y, K417N, N440K, N501Y, N679K, N764K, N969K, P681H, Q493R, Q498R, Q954H, S373P, S375F, S477. The H655Y, N679K, and P681H mutations are expected to accelerate spike cleavage and viral transmission, whereas the N501Y and Q498R variants are thought to improve ACE2 receptor binding [19]. Although BA.4/5 is closely related to BA.2, it also has a modification at RBDL452R found in the Delta form, F486V, and the wild-type amino acid at Q493, in addition to the 69–70 ablations in the nonterminal region observed in Alpha, BA.1, and BA.3. All previously found changes in Alpha (N501Y), Beta (K417N, E484K, N501Y), Gamma (K417T, E484K, N501Y), and Delta (L452R, T478K) have been assembled in the VOC with the exception of E484A in BA.4/5 as opposed to E484K in Beta and Gamma [20].

*1.2. Transmissibility of Omicron Variant*

The review of early South African data revealed that the Omicron type is capable of spreading much more quickly from person to person. There were significant differences in infection rates between the Gamma and Beta pseudo viruses when compared to the wild type when linear regressions were used. Delta was nearly twice as efficient as Gamma in invading specific cell types. The Omicron variation had infection rates four times greater than the wild type, while the Delta version had disease prevalence twice as high. For the Omicron variation, the ACE2-mediated infection was more efficient than the wild-type strain or another mutant strain, as shown by these data [10]. The Omicron variant's high transmittance may be influenced by a variety of circumstances. The Omicron variety of SARS-CoV-2 contains over 30 alterations in the genomic architecture of its spike protein, which is used to identify host cells, according to genome sequencing data [21]. Increased transmission of the Omicron variation is facilitated by the N501Y and Q498R mutations, which both improve the binding affinity for the ACE2 receptor, a crucial factor in increased transmission. Furthermore, the greater transmissibility of the Omicron form indicates a high likelihood of reinfection in those who have already been infected with COVID-19. The 15 Omicron variant mutations H655Y and N679K identified at the fur in cleavage site (FCS) may promote spike cleavage and increase viral infectiousness. P681H, on the other hand, may increase transmissibility by boosting the cleavage of the spike proteins [22]. It was hypothesized that the Omicron SARS-CoV-2 version had a greater affinity for ACE2 than the Delta form. The greater affinity of Omicron variants to the ACE2 is due to several mutations in the receptor-binding domain of spike protein, such as N501Y, Q493R, Q498R, S371L, S375F, S373P, and T478K.21 through 32; consequently, this shows that omicron

VOC is more easily transmitted than other versions of the virus [23]. More specifically, the R493 and R498 regions of the Omicron RBD, as well as the D30/E35 and D38 sites of the hACE2, were shown to have large electrically charged contacts (salt bridges) and hydrogen bonds. S496 and H505 are two more mutant amino acids in the Omicron RBD that formed hydrogen bonds with the hACE2 binding site. Furthermore, the api-stacking attraction was seen across two tyrosine regions in the active site (RBD-Tyr501: hACE2-Tyr41), considerably lowering free energies and demonstrating that it is one of the key links sustaining the complex's stability [15]. Omicron, on the other hand, was less successful than Delta in promoting cell-to-cell fusion once inside the cells.

Cell culture findings indicate that the breakdown of the s2 region at the spike's protein is comparatively reduced in the omicron genotype compared to the Delta genotype, further elaborating the route of viral administration of the omicron genotype to be a typical end ocytic route as compared to the cell fusion mechanism used by previous variants, particularly the Delta variant [24–26].

A common finding in tissues removed from the lungs following a severe sickness is the presence of fusion cells. It was discovered that the Omicron version was much weaker at replicating than the Delta variant in a spreading infection experiment using lung cells, confirming the results of a reduced entry [27].

Virulence of Omicron

Several researchers have reported on the increasing dissemination of the omicron genotype and its origin, and its etiology has been explored in the previous section. This section will go through the pathogenicity of omicron in great depth. Observational studies have shown pathogenicity in terms of its capacity to cause severe diseases that result in hospitalization or death. The pathogenicity of the omicron genotype varies by country. According to early statistics combined with clinical data, the fourth and fifth waves of the omicron version have lowered death and hospitalization rates in South Africa, particularly in Gauteng province [28,29]. In accordance with this, during the Omicron outbreak, Canada, Brazil, India, and Pakistan had lower hospitalization rates and daily mortality.

However, the Omicron variation influx was linked to more frequent regular episodes but less severe COVID-19 ailments [20,30,31].

According to the researchers, Omicron is less virulent than previous versions, notably Delta [32]. In contrast, during the Omicron pandemic, Australia, the United Kingdom, the United States, and France had greater mortality and hospitalization, with one to three times the daily cases than the Delta strain. The Omicron variation, on the other hand, had less of an impact on the daily ICU cases. This was most likely attributable to the overall number of vaccinated persons in each nation, as well as various preventions in each country and other variables [20,33]. Children accounted for 18% of COVID-19 hospitalised patients. Clinical signs were noted in 138 young patients, the majority of whom were under the age of four, in South Africa's Gauteng province. Fever, cough, shortness of breath, convulsions, vomiting, and diarrhea were among the symptoms [34]. When compared to the Delta wave, the Omicron infection wave showed a greater prevalence of paediatric infections necessitating hospitalization in the United States (5 percent of hospitalizations involving children) and the United Kingdom (particularly in kids under one year old) [35]. Owing to Omicron's immunological escape, it can infect people who have previously developed a defense against previous SARS-CoV-2 infections. Although it is possible to anticipate that recontamination or infestation of individuals with innate immunity that has been boosted by vaccination would certainly result in less severe symptoms, this does not generally entail that Omicron is found to be mentally less harmful [36].

The declined virulence of the Omicron variant is corroborated further in preclinical investigations by comparing it to earlier variations. In mouse models, the wild-type and Delta genotype multiplied to increasing copy numbers across the pulmonary system, whereas omicron had 1000 times less viral duplicate counts and 50 times lower active viral titers in both the nasal tract and the lungs [26]. Omricon virulence is reduced in hamsters as well, with a considerably lesser time reduction (10 folds) in the Omicron genotype [37].

Furthermore, Omicron-infected hamsters had less inflammation and hyperplasia in their lungs than Delta-infected hamsters [25].

The Emergence of Novel Omicron Variants

The fourth COVID-19 spike was predominantly driven by three previous origins of the SARS-CoV-2 Omicron mutant (BA.1, BA.2, and BA.3) [21,38]. BA.4 and BA.5 had swiftly supplanted BA.2, accounting for more than half of all sequenced cases in South Africa by the first week of April 2022. Nonetheless, BA.4 has been discovered in Austria, the United Kingdom, the United States, and Denmark, while BA.5 has been discovered in Germany, Portugal, the United Kingdom, and the United States. As a result of the projected increased disease transmission, these alterations may result in a significant overall increase in COVID-19 cases in the near future. These two different genotypes (BA.4 and BA.5) have just been classed as variants of concern (VOC) [39]. The continual discovery of heritably unique Omicron progenitors lends credence to the notion that a particular repository, such as persistent infections in humans and/or animal carriers, may be impacting the virus's ongoing evolution and dissemination [40]. In comparison to BA.1 and BA.2, BA.4/5 demonstrates reduced neutralisation by sera from persons who got AstraZeneca or Pfizer vaccine injections. Furthermore, when sera from outbreak infections induced by the BA.1 vaccine are utilized, the neutralization of BA.4/5 falls noticeably, increasing the danger of repeated Omicron infestations [20].

Meanwhile, Pakistan is coping with the fifth wave caused by the new kind, Omicron (or B.1.1.529). It is now the most prevalent strain in Pakistan, notably in Karachi, where the positive rate exceeds 40%. Furthermore, the identification of highly transmissible BA.2 mutations in Islamabad, Pakistan, may increase the number of positive cases [41]. People of all ages, typically more men, both vaccinated and unvaccinated, are afflicted, and re-infection is occurring. Since vaccination recipients do not need to be hospitalised, ICUs are only used for extreme cases. Fortunately, there is a lower mortality rate than when compared to the Delta virus. We still do not know or comprehend a lot of issues. The problem that will undoubtedly develop is that a sizable portion of the earth will become infected if Omicron continues to grow like a disaster. Only a small percentage of people with pathologies and those who have not had vaccines and are, thus, unable to fight off the virus may become really ill. These people require hospitalization and intensive care therapy. This might make the medical system totally unresponsive and impose a heavy burden on medical institutions, particularly in developing countries including Pakistan. This may be viewed in the USA and the UK as a result of the spread of Omicron [42].

*1.3. Economic Loss Due to Omicron*

The third wave of COVID has hit the world with the Omicron variant. There have been various indirect effects of this wave, such as travel bans, education institutes getting closed, and lockdowns, and one of the major effects includes economic loss. During crises like these, it becomes imperative that worldwide governments take rational decisions.

When Omicron emerged, WHO suggested not implementing travel bans instead of using risk-based and scientific approaches. After this was announced and identified from where the new variant originated, the travel ban was ordered on all those countries. This included various travel bans in African countries. To protect public health, strict measures were taken like furlough, quarantine, and self-isolation, which were meant to maintain food security, shelter, and livelihood. The world's economy has been seriously affected by this third wave. Economic activities have been halted, and peace of mind and body has been disturbed devastatingly [43]. Experts are now trying to focus on mass immunization in addition to public health protection measures. Health care infrastructure also needs continuous funding to be stronger for current present variants and also any new that may emerge. This has resulted in a great financial loss for economies around the world [44]. Due to the quality of Omicron having a high transmission rate that is increasing the infected individual's ratio rapidly, the hospital staff and capacity are running short. It is feared

that if hospital health care systems are overwhelmed, it may increase the mortality rate devastatingly. In the times that Omicron has struck us, we will either come up with a strategy to overcome this or learn how to 'live with it' [45].

## 2. Impact on Pakistan

Pakistan is a country with a developing status and has a population of about 220 million people. Gilgit Baltistan, Punjab, Khyber Pakhtunkhwa (KPK), Sindh, and Balochistan are the five provinces that make up Pakistan. According to the Pakistani Ministry of Health, the country's first confirmed incident of COVID-19 happened on 26 February 2020, and since that time, a constant spread of the disease has been observed throughout the country [46].

The data in (Figure 2) from 26 February 2021 to 9 June 2022. The pie chart depicts the number of positive cases along with the percentage of deaths from COVID-19 in Pakistan's provinces. It indicates that the highest percentage of cases are found in Sindh, followed by Punjab, and the lowest percentage is found in Gilgit Baltistan.

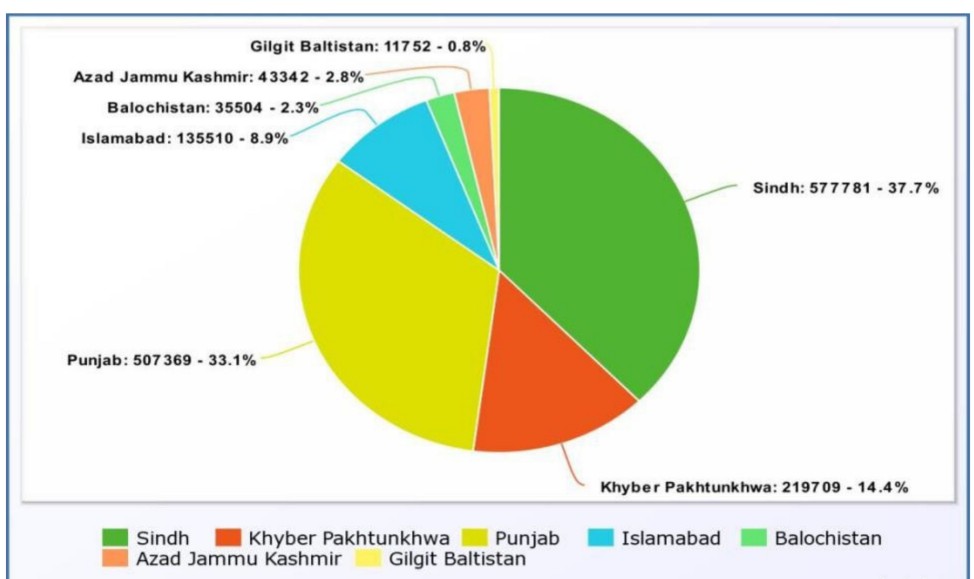

**Figure 2.** Provincial positive cases [47].

The percentage of deaths in (Figure 3) from 26 January 2020 to 9 June 2022 in each province is represented by the pie chart above. Punjab has the highest death rate in the country, at 44.5 percent, while Sindh has a mortality rate of 26.9 percent, according to the data. The death rate in the federal territory, which includes Islamabad, is estimated to be 3.4 percent. Gilgit Baltistan has the lowest death rate at 0.6 percent.

The pie chart in (Figure 4) above depicts the number of persons who survived in each province from 26 February 2020 to 9 June 2022. When the recovered incidents are compared to the confirmed cases in each province, it is shown that Sindh had 577,781 positive cases and 567,237 recoveries, whereas Punjab had 507,367 cases and 491,786 fatalities. Furthermore, the capital city of Islamabad has 135,510 cases and 134,140 survivors, according to the data. The lowest recovery number is 11,546 in Gilgit Baltistan, which is because the lowest number of positive cases was 11,752.

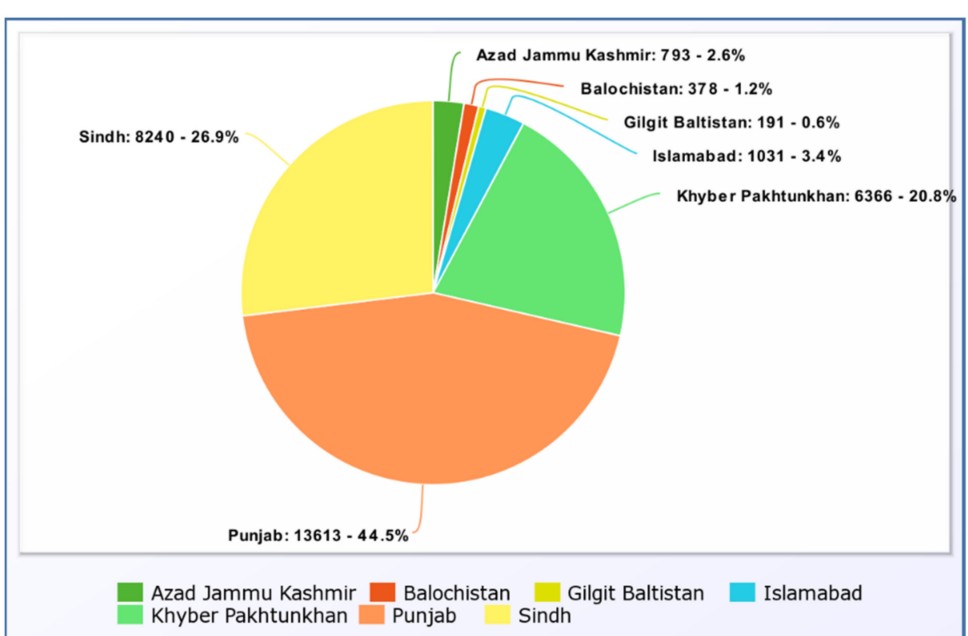

**Figure 3.** Provincial Death Rate in number and percentage [47].

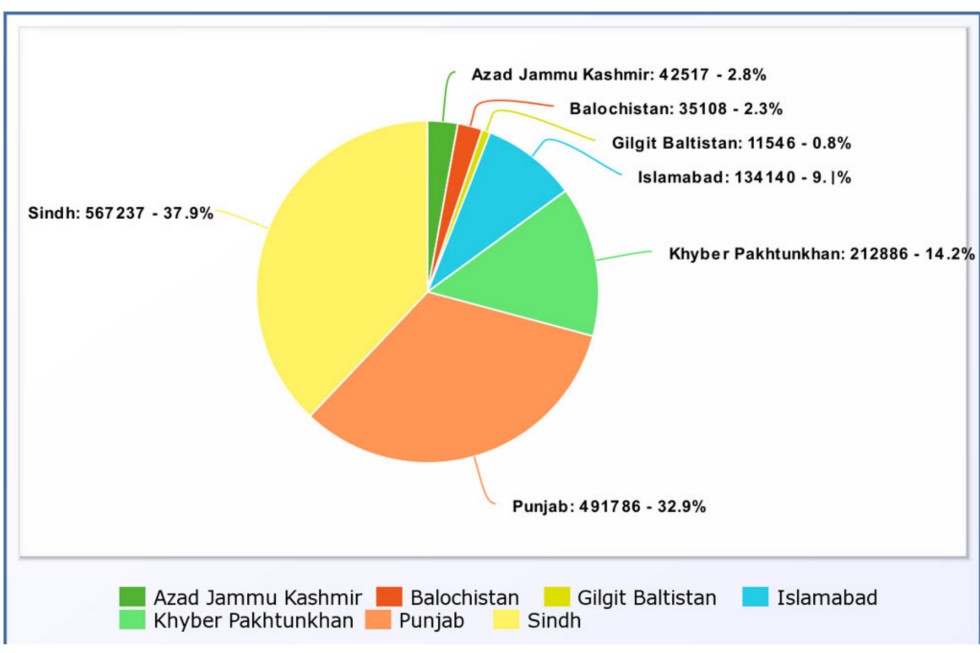

**Figure 4.** Provincial Recoveries [47].

## 3. Control and Prevention

The established public health prevention methods such as mask use, physical separation, avoidance of confined areas, preference for outdoor activities, and hand cleanliness have been proven effective against previous variants (Figure 5). Therefore, they should be equally effective against the Omicron variant [46].

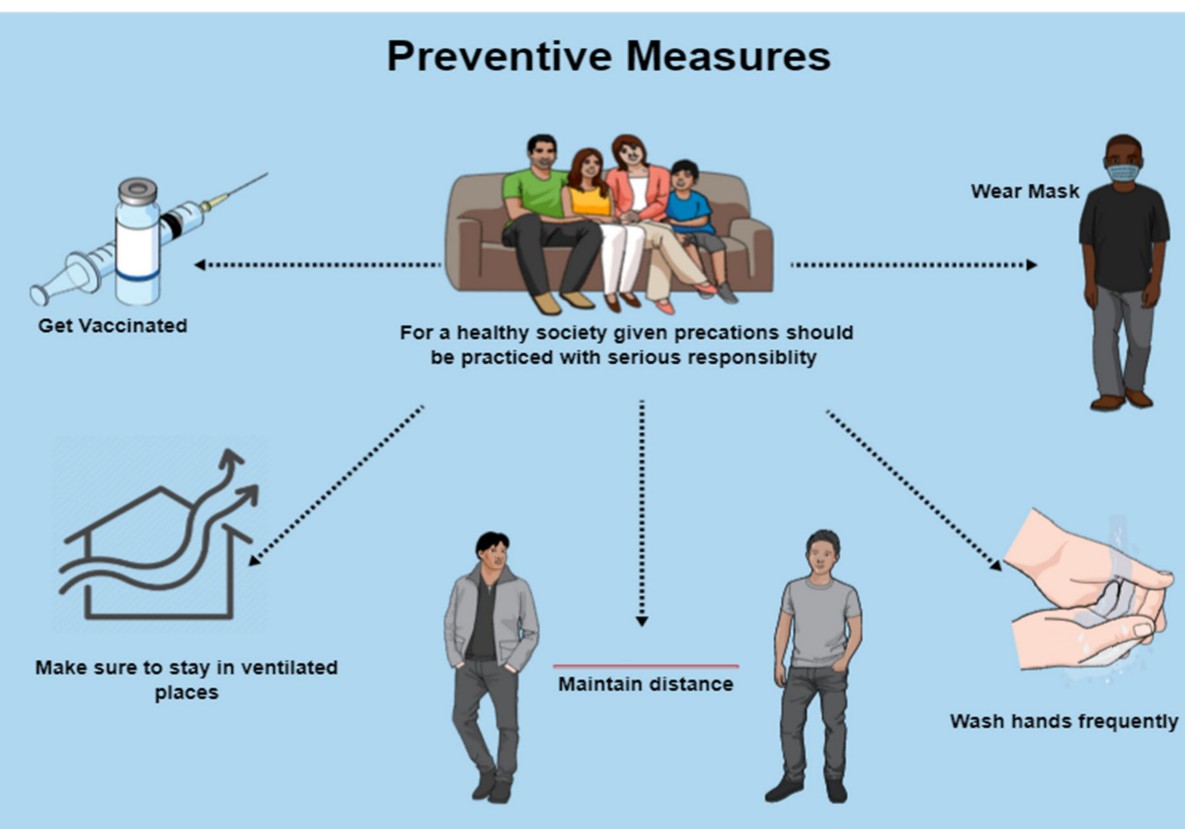

**Figure 5.** Preventive measures for COVID-19 and its variants.

Although COVID-19 vaccinations have become less effective towards mutant virus infections, they nonetheless aid in the prevention of serious diseases, hospitalization, and fatalities. Poor antibody titers in SARS-CoV-2 patients who have been infected or immunized may help in the emergence and development of novel variations. Recent investigations have shown that giving an additional boosting dose to a succeeding inoculation will preserve, if not increase, vaccine effectiveness. The COVID-19 vaccine with an extra dose six months after initial vaccination could help limit Omicron infection and spread (Figure 6) [48].

It is well understood that the variants of concern are more harmful and have more potential to cause disease with rapid spreading capability. The most obvious prevention is to get tested on regular basis along with immediate vaccination and booster administration. For the broader understanding and strategy development of the control and prevention of Omicron, its biological aspects, virology, therapies, clinical phenomena, and dynamics of alternations in the genome should be studied and researched thoroughly. Translational medication, along with clinical assistance, can prove to be quite helpful for a better understanding of disease severity, therapeutic and diagnostic failure, immune escape, or diversity of the infection that helps in the prevention of disease [49].

Above all, the basic preventive measure to control Omicron infection is the avoid its transmissibility. The best solution for this is to practice preventive measures published by the world health organization including maintenance of social distancing, periodic hand washing, and wearing masks. It has been surprisingly effective for the wild-type coronavirus so it can do wonders for Omicron as well. Secondly, the accuracy in diagnosis should not be compromised because the PCR tests based on the spike gene identification have failed to a greater extent in terms of accuracy; therefore, the diagnostic accuracy needs to be improved [48].

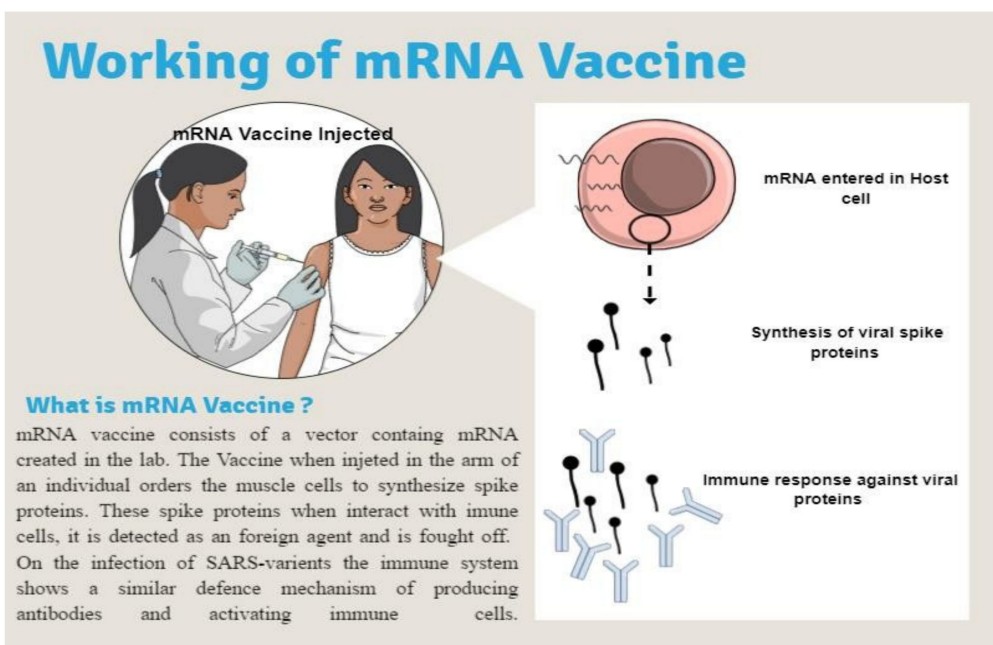

**Figure 6.** Working of mRNA Vaccine [50].

It is projected that a combination prevention strategy involving vaccination and public health initiatives will continue to be an effective method [46].

*3.1. Vaccination*

Vaccines are vital assets for the preventative measures, containment, and/or eradication of infectious illnesses and are fundamental to community healthcare systems across the world (Table 1) [51]. The development and distribution of efficient Coronavirus disease 2019 (COVID-19) vaccines highlighted a notable success across the continuing pandemic. As shown in Table 1, many parallel vaccine engineering initiatives were launched to lessen the effects of the COVID-19 epidemic, which was caused by the previously discovered virus known as severe acute respiratory syndrome Coronavirus 2. (SARS-CoV-2). Out of 339 vaccine exploratory investigations, 139 vaccine prototypes have reached the trial stage, according to the World Health Organization's (WHO) vaccination monitoring system [52].

**Table 1.** Characteristics of vaccine.

| Vaccine Type | Vaccine Name/Sponsor | Efficacy | Approval Status | Dosing Regimen | Storage |
|---|---|---|---|---|---|
| mRNA | BNT16b2 (Pfizer/BioNTech) | 95% | Approved for ≥16 yrs in 84 countries | 2 dosages with gap of 21 days | −60 to −80 °C; 2 to 8 °C for a single month |
| mRNA | mRNA-1273 (Moderna) | 94% | Approved for ≥16 yrs in 46 countries | 2 dosages with gap of 28 days | −50 to −15 °C; 2 to 25 °C for a single month |
| mRNA | CVnCoV(CureVac) | 47% | Did not meet the approval criteria/Withdrawal from Phase 3 trials | 2 dosages with gap of 28 days | 5–60 °C |
| Human adenovirus vectors (rAd26 and rAd5) | Gamaleya National Research Centre for Epidemiology and Microbiology | 92% | Approved for ≥18 yrs in 68 countries | 2 dosages with gap of 1–3 months | 2–8 °C |

**Table 1.** *Cont.*

| Vaccine Type | Vaccine Name/Sponsor | Efficacy | Approval Status | Dosing Regimen | Storage |
|---|---|---|---|---|---|
| Chimpanzee adenovirus vector | ZD1222 (AstraZeneca/Oxford) | 70% | Approved for ≥18 yrs in 98 countries | 3 dosages with gap of 21 days | 2–8 °C |
| Human adenovirus vector | Ad26. COV2.S (Johnson & Johnson) Janssen | 67% | Approved for ≥18 yrs in 41 countries | single dose | 20 °C, 2–8 °C; 3 months |
| Human adenovirus vector | CanSino | 66% | Approved for ≥18 yrs in 5 countries | single dose | 2–8 °C |
| Inactivated vaccine based on the CZ02 strain; aluminium hydroxide-adjuvanted | CoronaVacSinovac/China National Pharmaceutical Group | 50.65% to 83.5% | Approved for ≥18 yrs in 3 countries (Brazil, Indonesia, and Turkey) | 2 dosages with gap of 14 days | 2–8 °C/5 months 25 °C/42 days 37 °C/28 days |
| Inactivated vaccine based on the 19nCOV-CDC-TAN-HB02 strain; aluminium hydroxide-adjuvanted | BBIBP-CorV (Sinopharm/Beijing Institute of Biological Products) | 78.1 | Approved for ≥18 yrs in 4 countries | 2 dosages with gap of 21 days | 2–8 °C |
| Whole-virion inactivated SARS-CoV-2 vaccine formulated with a toll-like receptor 7/8 agonist molecule (IMDG) adsorbed to alum (Algel) | Covaxin/Bharat Biotech | 78% | Approved for ≥18 yrs in India | 2 dosages with gap of 28 days | 37 °C/7 day |
| Multitope peptide-based spike protein S1 subunit RBD CpG1 And aluminum | UB-612 (COVAXX and United Biomedical Inc.) | N/A | Withdrawn from trials | N/A | 2–8 °C |
| Dimeric form of RBD adjuvant with aluminium hydroxide | ZF2001 (Chinese Academy of science) | N/A | N/A | NA | 2–8 °C |
| Recombinant protein (insect cell) + matrix M adjuvant | Novavax | 90.4% | Approved for ≥16 yrs in 84 countries | 2 dosages with gap of 21 days | 2–8 °C |
| Recombinant protein (insect cell) + AS03 adjuvant | Sanofi/GSK | N/A | N/A | 2 dosages with gap of 21 days | 2–8 °C |
| Protein subunit vaccine | EpiVacCorona (Vector Institute) | N/A | Early use in Russia | 2 dosages with gap of 21 days | 2–8 °C |

*3.2. Vaccine Efficacy*

The effectiveness of the vaccine against the variant varies. Studies have shown that all currently available vaccines have lower efficacy against the various variants emerging compared to the original strain. The efficacy of Astra Zeneca (AZD1222) fighting the Alpha variant was 74% which dropped to 67% for the Delta variant. Just like that, the effectiveness of Pfizer (BNT162b2) was 94% against Alpha but 88% for Delta. The severity

of the disease, and hospitalizing the patient, get greatly reduced by taking double doses of both vaccines [53].

Experimental studies show that neutralization of antibody titers done by Pfizer was 44-fold lower and by Astra Zeneca was 34-fold lower. Just like that, all the other vaccines such as the Sinopharm, Moderna, and Sputnik also showed lower neutralizing antibody titers against Omicron. Emerging data from the UK demonstrated how, although double doses of vaccines did not help with symptomatic Omicron, it reduced the chances of patient hospitalization. This is suggesting that cell-mediated immunity is at least partially retained against Omicron [53].

Although, COVID-19 vaccines such as Pfizer, Moderna, Sputnik, Astra Zeneca, Can Sino, Sinopharm, and others aim to protect the health and well-being of people from the deadly virus [54]. However, their safety and efficacy have been highly debated among different scholars [55–57]. Among different types of vaccines, mRNA vaccines have better efficacy and safety profiles for two reasons. Firstly, these vaccines do not contain live or inactive viruses. Secondly, they do not alter the DNA of the host cell [58]. mRNA vaccines benefit vaccinated individuals by protecting them from ailments like COVID-19 without compromising the possible serious side effects of the disease [59]. Traditional vaccinations, such as live attenuated virus, inactivated virus, and protein subunit vaccines, have several drawbacks, but mRNA vaccines offer several advantages over them [60]. In prior clinical studies with mRNA vaccine candidates against the HIV, rabies virus, and cancers, the safety and immunogenicity of this platform were also shown [61]. As a consequence, several academic institutions and corporations have adopted this platform to develop COVID-19 vaccines. Numerous mRNA vaccines have undergone the most advanced clinical trials. In recent phase III clinical studies, Moderna/NIAID and BioNTech/Pfizer have shown that their mRNA vaccines against COVID-19 were safe and protected against >90% of cases; therefore, these vaccines have been licensed for use in particular countries. The COVID-19 mRNA vaccine has significant benefits over competing vaccines. These include a fast development cycle, simplicity of industrialization, a simple manufacturing procedure, the capacity to adapt to novel variations, and a greater degree of immune response induction [62].

Boosters were introduced way before Omicron came in. However, they have been a hot topic of debate over the case of its efficacy ever since. Many countries have been giving boosters following the vaccinations provided. The serum of individuals who have been given boosters has shown an increasing neutralization of antibody tiers against Omicron. When giving mRNA booster with a vaccine, the efficacy increased by 5% to 73% in AstraZeneca vaccinated individuals and a 19% increase to 77% with Pfizer. By that time, evidence in favor is emerging prominently, and boosters are being administered increasingly in all countries. Studies have shown that a heterologous booster has been more helpful than a homologous booster. Countries that can afford to are promoting and providing booster doses as a trusted resort to severe Omicron [53].

By 7 January 2022, there have been a very limited number of countries that have rising cases of Omicron. Studies show how the efficacy of the vaccine has increased, especially in double-dose vaccines with the heterologous booster. However, the efficacy is more of an advantage against transmission and does not guarantee any kind of protection against serious disease [63].

During the first week of February 2021, Pakistan began to administrate the COVID-19 vaccination program throughout the country [64].

The Pakistan vaccination status graph provides the vaccination stats until 3 March 2022 (Figure 7). In a population of 220 million people, 992,921,29 people have received their whole vaccination schedule. The initial dose, on the other hand, is provided to 127,117,340 persons. Since the booster dose program has only recently begun, only 451,2786 shots have been provided. At the same time, the total number of dosages administered is 215,539,999.

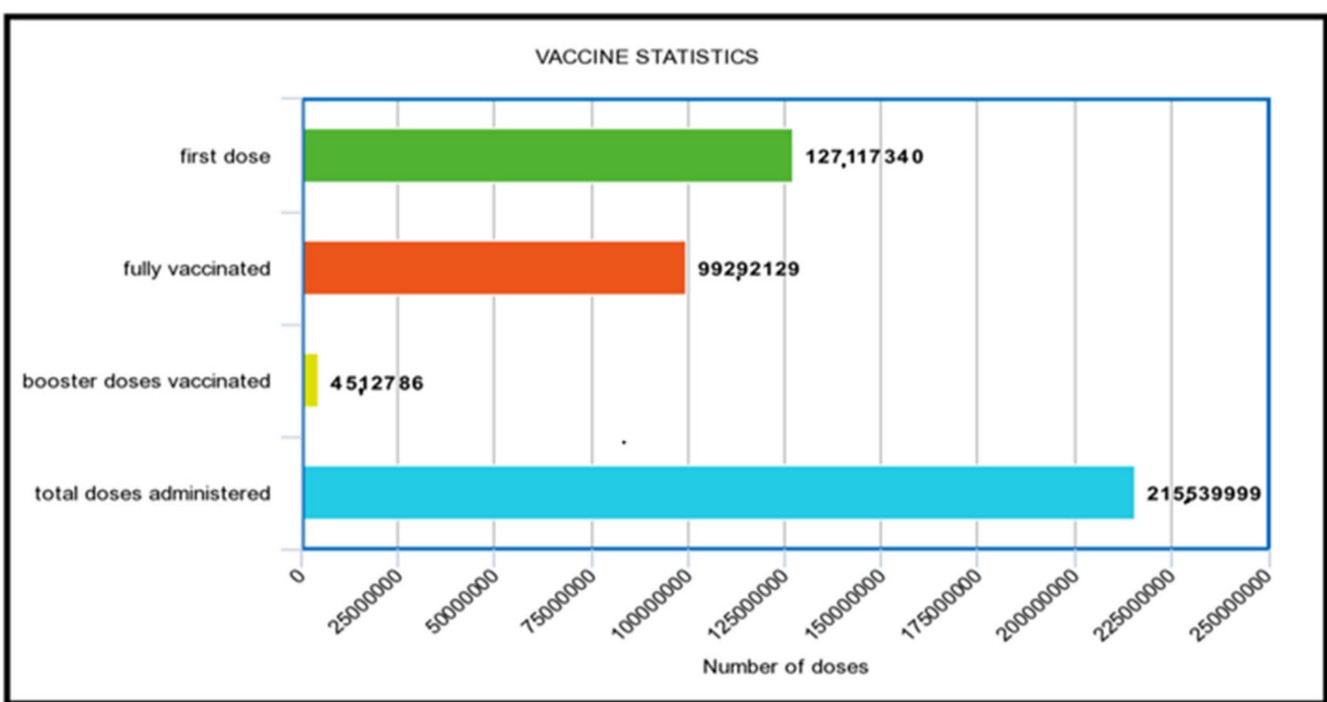

**Figure 7.** Pakistan vaccination status [47].

*3.3. Diagnostic Test*

Diagnostic testing that is timely and reliable is an essential component of the public health response to Coronavirus. The use of antigen tests is widely used in many countries because they allow point-of-care testing that is quick, inexpensive, and easily accessible.

Nucleocapsid protein is detected by the great majority of antigen testing; it is a structural protein that shows less variation in comparison to the spike protein across distinct CoV-2 lineages, which results in severe acute respiratory syndrome (SARS). Even though antigen tests are not as sensitive as reverse transcription-PCR (RT-PCR) testing, their capacity to swiftly discover patients with high viral loads has clinical and public health applications in a variety of settings [65].

Highly conserved regions are frequently used as PCR targets to avoid PCR test failure; however, these can be difficult to locate for novel diseases. Multiple PCR targets can be utilized within a single test to compensate for single target failures, reducing this risk. Moreover, the diagnostic performance of available PCR assays is currently unknown, as all information accessible at this early stage is based on manufacturers' in silico analyses [66].

To quickly analyze the effectiveness of the existing systems, laboratories should share RNA samples from positive cases. Furthermore, political expectations for widespread surveillance of the Omicron variant's propagation are impractical at this time, given that laboratories are already under strain owing to the ongoing and strong Delta wave, which requires enormous volumes of PCR testing. There should be a targeted approach with a continuation of the national surveillance program for the appearance of variations using whole-genome sequencing. This program should be continued for the next 1–2 years to give enough sequencing information for vaccine development and diagnostic assay adaption [66].

**4. Treatment**

Medications suggested by the National Institute of Health (NIH) for non-hospitalized cases (Table 2) [67].

**Table 2.** Treatment [68].

| Drugs | Doses | Recommended Population |
|---|---|---|
| Ritonavir boosted Nirmatrelvir (PAXLOVID) | Oral intake of Nirmatrelvir 300 mg with ritonavir 100 mg for 5 days | Adult outpatients older than 18 years old who have nonhypoxic COVID-19 and are at high risk of severe disease progression (e.g., due to advanced age, comorbidities, not having been vaccinated, or being immunosuppressed) are candidates for treatment |
| Sotrovimab | IV infusion once in the initial ten days of symptoms | Approved for the treatment of COVID-19 in adolescents (aged less than 12 years and weighing nearly40 kg) and adults who do not require oxygen supplementation but who are at a high risk of advancing to severe COVID-19 |
| Remdesivir (VEKLURY) | Within seven days of symptoms emergence, 200 mg IV infusion on Day one and 100 mg infusion on 2nd and 3rd day. | Adults and children over the age of 12 who weigh a minimum of up to 40 kilos and are hospitalized can administer this medication |
| Molnupiravir | Oral intake of 800 mg for five days on the initial five days of symptoms | It is predicted that molnupiravir will be effective against the Omicron VOC |
| **Treatment for immunocompromised and patients with medical contraindications** [68] | | |
| Tixagevimab and Cilgavimab (EVUSHELD) | Intramuscular injection once of 150 mg Tixagevimab and 150 mg cilgavimab | N/A |

## 5. Conclusions

Omicron has become a concerning variant of COVID-19. Much more work has to be done in understanding the new variant for improved treatment and vaccine development. For now, preventive measures similar to COVID-19 should be followed with vaccination.

In a developing country like Pakistan, which is already dealing with major issues connected to security, the economy, political instability, epidemics, and pandemics pose a serious threat to the country's already precarious healthcare system. Political inertia, inefficient health policies, ineffective governance, and an indifferent attitude on the part of the people regarding general preventative measures are all contributing to the situation's deterioration. We have reached a critical juncture at which the country must reevaluate its goals and strive toward strengthening the health system to successfully handle health concerns [64].

Inexpensive and accurate diagnostic tests are required for the detection of strain with its treatment. However, the treatment used at present is similar to that of the first wave but it is effective. Studies on Omicron conserved regions could lead to an efficient methodology for variant elimination.

**Author Contributions:** Conceptualization, S.K. (Sarmir Khan); methodology S.H.K.; F.H. and J.M.; Softwere, S.H.K.; F.H. and J.M.; formal analysis, F.K.; I.K. and M.A.Z.; visualization, S.H.K.; F.H. and J.M.; investigation, S.K. (Sarmir Khan); S.H.K.; F.H. and J.M.; Data collection, S.K. (Sarmir Khan), S.H.K., F.H., J.M.; Writing-original draft preparation, S.K. (Sarmir Khan); S.H.K., F.H.; J.M.; I.K.; F.K. and M.A.Z.; Writing-review and editing, S.K. (Sarmir Khan); I.U.; M.A.Z. and S.K. (Shazia Kousar); supervision, S.K. (Sarmir Khan); I.U. and S.K. (Shazia Kousar). All authors have read and agreed to the published version of the manuscript.

**Funding:** This research received no external funding.

**Institutional Review Board Statement:** Not applicable.

**Informed Consent Statement:** Not applicable.

**Conflicts of Interest:** The authors declared no conflict of interest.

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
