# Peer review of "The Burden of Omicron Variant in Pakistan: An Updated Review"

_covid, doi:10.3390/covid2100105_

Round 1
Reviewer 1 Report
I read the review entitled "The burden of Omicron variant in Pakistan: updated review". The key messages from the review are missing. The title is an updated review, but the data is a bit old. Need to describe the key messages at the end of the manuscript in a conclusion and a clear recommendation for further research. Some specific comments are given below:
- Change the Title "The burden of Omicron variantin Pakistan: updated review: and make it "The burden of Omicron variant in Pakistan: an updated review"
-Update Figures 1, 2, and 3 with newly updated data
-Figure 4 is unnecessary, What is the justification of use this figure in the manuscript. If you are interested to keep it, update it with recent data
-I see only 1 reference in 4.1, 4.2, 4.3, and 4.4. It is not logical to use a part of a manuscript with 1 reference. Use references and cross-references to describe this part
-Describe the control and prevention measures appropriately
Reviewer 2 Report
The authors in this review attempted to cover recent development to tackle the need-of-the-hour COVID19 pandemic. In this reference the authors tried to cover the latest Omicron variant with regard to its epidemiology, transmissibility, available vaccine and drug options and its impact on the economy of Pakistan. The review is describing an important role that vaccines can have in controlling the pandemic, however, there are a lot of caveats to the literature interpretation and presentation. The review needs substantial amendments to be considered for publication. I have following comments to improve the manuscript:
- The content of the review does not justify the title of the manuscript completely and only a very small part focus on Omicron variant and its virulence.
- Another concern regarding the review is related to updated information that authors want to convey and level of novelty compared to the plethora of reviews already published on similar topic within last few months (https://onlinelibrary.wiley.com/doi/epdf/10.1111/joim.13478; https://onlinelibrary.wiley.com/doi/epdf/10.1002/jmv.27697; https://www.thelancet.com/journals/lanres/article/PIIS2213-2600(21)00559-2/fulltext). Considering these reviews, this manuscript does not add much value on Omicron than what is already known to most people.
- Not all recent variants are covered. The variants Kappa, epsilon, Iota and Mu are also recent one and not even mentioned in the manuscript (or Figure 5 which is actually the Figure 1 in manuscript probably typing error). I will recommend to add comparative information of these variants against Omicron. The information about variants can be presented as phylogentic trees for easy understanding to the reader.
- Detailed information about comparative virulence of SARS-CoV2 variants should improve the manuscript novelty.
- Lot of vaccines which are currently in use (globally or region specific) are not mentioned in the manuscript and information provided related to handful of vaccine (moderna, Pfizer, astrazenca) and is not covered in detail. The mode of action for vaccine should be discussed including the efficacy, mode of inhibition for FDA approved vaccines (only mRNA vaccine was disussed). Why mRNA vaccines are promoted than viral antigenic protein based vaccines should also be discussed.
- Most of the topics are superficially touched and few details were mentioned (not up to date). I miss details on genomic characterization and virion structure, replication and pathogenesis, Comparative virulence of SARS CoV2 variants, transmission etc.
- Probably because of writing style but the information provided in this review is very limited and covering the information from 2019 (the very beginning of pandemic) that too is related to general viral prevention practices followed in lab or public places.
- The writing style could be improved and structure of review can be improved using more tables, figures showing comparison of different approaches employed to preven viral transmission and control.
- The literature is not up to data and key publications related to this topic are not cited. Please perform a thorough literature search making further changes to this manuscript. I specifically miss approporiate citations at many places. For example, Line 27-29, Line 193-195, Line 218 to 227 etc.

Round 2
Reviewer 1 Report
Visiable improvement of manuscript done by the authors in revised version. I think language is still problemetic. After, language corrrection, the manuscript can be accepted.
Author Response
Response to Reviewer 1 Comments
Point 1: Change the Title "The burden of Omicron variant in Pakistan: updated review: and make it "The burden of Omicron variant in Pakistan: an updated review"
Response 1: The title has been improved to "The burden of Omicron variant in Pakistan: an update”
Point 2: Update Figures 1, 2, and 3 with newly updated data
Response 2: Pie charts have been updated; they show the data from 26 February 2020 to 9 June 2022.
Point 3: Figure 4 is unnecessary, what is the justification of use this figure in the manuscript. If you are interested to keep it, update it with recent data
Response 3: Figure 4 has been deleted due to irrelevant data.
Point 4: I see only 1 reference in 4.1, 4.2, 4.3, and 4.4. It is not logical to use a part of a manuscript with 1 reference. Use references and cross-references to describe this part
Response 4: Heading 4.1 to 4.4 has been updated. More references have been added in each heading.
Point 5: Describe the control and prevention measures appropriately
Response 5: Control and Prevention heading has been more elaborated.
Reviewer 2 Report
The authors have not provided response to any of the questions raised by me in my previous review in their rebuttal letter. The rebuttal letter that I downloaded from the MDPI website is non-scientific and was written in a format that I have never seen in any of my previous review. As a reviewer I cannot track if authors have addressed all the concerns raised by me to my satisfaction.
Therefore, all my previous comments sustain and I ask authors first to write an appropriate rebuttal letter before I check for the amendments.
Author Response
Point 1: The content of the review does not justify the title of the manuscript completely and
only a very small part focus on Omicron variant and its virulence.
Response 1: The title has been improved to "The burden of Omicron variant in Pakistan: an update”. We have written the content accordingly.
Point 2: Another concern reagrding the review is related to updated information that authors
want to convey and level of novelty compared to the plethora of reviews already
published on similar topic within last few months
(https://onlinelibrary.wiley.com/doi/epdf/10.1111/joim.13478;
https://onlinelibrary.wiley.com/doi/epdf/10.1002/jmv.27697;
https://www.thelancet.com/journals/lanres/article/PIIS2213-2600(21)00559-
2/fulltext). Considering these reviews, this manuscript does not add much value on
Omicron than what is already known to most people.
Response 2: Added more information related the title.
Point 3: Not all recent variants are covered. The variants Kappa, epsilon, Iota and Mu are also
recent one and not even mentioned in the manuscript (or Figure 5 which is actually
the Figure 1 in manuscript probably typing error). I will recommend to add
comparative information of these variants against Omicron. The information about
variants can be presented as phylogentic trees for easy understanding to the reader.
Response 3: Virulence and pathogenesis had been discussed
Point 4: Detailed information about comparative virulence of SARS-CoV2 variants should
improve the manuscript novelty.
Response 4: A diagram has been added to present the structure of virus and its virulence.Data about other variants (kappa, my etc).
Point 5: Lot of vaccines which are currently in use (globally or region specific) are not
mentioned in the manuscript and information provided related to handful of vaccine
(moderna, Pfizer, astrazenca) and is not covered in detail. The mode of action for
vaccine should be discussed including the efficacy, mode of inhibition for FDA
approved vaccines (only mRNA vaccine was disussed). Why mRNA vaccines are
promoted than viral antigenic protein based vaccines should also be discussed.
Response 5: mRNA vaccines and multiple companies such as Cansino has been discussed.
Point 6: Most of the topics are superficially touched and few details were mentioned (not up
to date). I miss details on genomic characterization and virion structure, replication
and pathogenesis, Comparative virulence of SARS CoV2 variants, transmission etc.
Response 6: All the headings have been more elaborated.
Point 7: Probably because of writing style but the information provided in this review is very
limited and covering the information from 2019 (the very beginning of pandemic) that
too is related to general viral prevention practices followed in lab or public places.
Response 7: Correctios have been made accordingly.
Point 8: The writing style could be improved and structure of review can be improved using
more tables, figures showing comparison of different approaches employed to preven
viral transmission and control.
Response 8: It has been corrected.
Point 9: The literature is not up to data and key publications related to this topic are not cited.
Please perform a thorough literature search making further changes to this
manuscript. I specifically miss approporiate citations at many places. For example,
Line 27-29, Line 193-195, Line 218 to 227 etc.
Response 9: Citations included.
Round 3
Reviewer 2 Report
I appreciate the effort made by the authors to make necessary amendments in response to my previous review. The manuscript looks much better compared to previous version. However still there are some caveats in review presentation that authors must work on before publishing in its current form. I am putting my comments under minor revision instead of major hoping that authors will make necessary changes with regard to figures, organization and updated content.
a) The review is very lengthy and covering lot of information which has been discussed in several reviews earlier. The information on vaccine, control prevention etc can be covered in tabular form to reduce the length.
b) The figures can be improved to include more information. The figures used in manuscript are not informative and resembles the cartoonish illustrations from the very beginning of the pandemic when there was not much detail available on COVID19 to make aware of public about pandemic.
c) I will recommend the authors to not to use the exact figures directly from others work. Figure 1 is directly from Lamers & Haagmans, 2022 and so does the figure 2 and others. The authors should use the information to create their own figures or chart but using the exact figure or chart would not be appropriate even if the authors cited the original source.
Author Response
Response to Minor Comments
Point 1: The review is very lengthy and covering lot of information which has been discussed in several reviews earlier. The information on vaccine, control prevention etc can be covered in tabular form to reduce the length.
Response 1: The information on vaccine, control prevention etc have been covered in tabular form
Point 2: The figures can be improved to include more information. The figures used in manuscript are not informative and resembles the cartoonish illustrations from the very beginning of the pandemic when there was not much detail available on COVID19 to make aware of public about pandemic.
Response 2: Corection has been done.
Point 3: I will recommend the authors to not to use the exact figures directly from others work. Figure 1 is directly from Lamers & Haagmans, 2022 and so does the figure 2 and others. The authors should use the information to create their own figures or chart but using the exact figure or chart would not be appropriate even if the authors cited the original source.
Response 3: created our own figures.
